# Sound Identification Method for Gas and Coal Dust Explosions Based on MLP

**DOI:** 10.3390/e25081184

**Published:** 2023-08-09

**Authors:** Xingchen Yu, Xiaowei Li

**Affiliations:** School of Artificial Intelligence, China University of Mining and Technology (Beijing), Beijing 100083, China

**Keywords:** gas and coal dust explosion, sound recognition, feature extraction, feature dimensionality reduction, MLP

## Abstract

To solve the problems of backward gas and coal dust explosion alarm technology and single monitoring means in coal mines, and to improve the accuracy of gas and coal dust explosion identification in coal mines, a sound identification method for gas and coal dust explosions based on MLP in coal mines is proposed, and the distributions of the mean value of the short-time energy, zero crossing rate, spectral centroid, spectral spread, roll-off, 16-dimensional time-frequency features, MFCC, GFCC, short-time Fourier coefficients of gas explosion sound, coal dust sound, and other underground sounds were analyzed. In order to select the most suitable feature vector to characterize the sound signal, the best feature extraction model of the Relief algorithm was established, and the cross-entropy distribution of the MLP model trained with the different numbers of feature values was analyzed. In order to further optimize the feature value selection, the recognition results of the recognition models trained with the different numbers of sound feature values were compared, and the first 35-dimensional feature values were finally determined as the feature vector to characterize the sound signal. The feature vectors are input into the MLP to establish the sound recognition model of coal mine gas and coal dust explosion. An analysis of the feature extraction, optimal feature extraction, model training, and time consumption for model recognition during the model establishment process shows that the proposed algorithm has high computational efficiency and meets the requirement of the real-time coal mine safety monitoring and alarm system. From the results of recognition experiments, the sound recognition algorithm can distinguish each kind of sound involved in the experiments more accurately. The average recognition rate, recall rate, and accuracy rate of the model can reach 95%, 95%, and 95.8%, respectively, which is obviously better than the comparison algorithm and can meet the requirements of coal mine gas and coal dust explosion sensing and alarming.

## 1. Introduction

Gas explosions, coal dust explosions, and gas and coal dust explosions (hereafter referred to as gas and coal dust explosions) are serious accidents that occur in coal mines [1,2,3,4], which will cause casualties and economic losses once they occur. Therefore, a large number of researchers have conducted a lot of research on them. The existing measures of monitoring are mainly realized by gas and temperature sensors, which are outdated monitoring means and have problems, such as slow reporting, a false alarm rate, and a high leakage rate. Therefore, it is of great theoretical significance and practical value to study the intelligent identification and alarms of coal mine gas and coal dust explosions.

Sound has the characteristics of a long propagation distance and small influence by bending and branching of the tunnel [3,4,5,6,7]; therefore, sound recognition has achieved better results in explosion identification in coal mines. The authors of [4] analyzed the characteristic differences between the sound of gas and coal dust explosions and other sounds in coal mines and proposed to realize the recognition of the sound of gas and coal dust explosions in coal mines by the method of sound recognition. The authors of [5] compared the decomposition results of coal mine underground sound through different EMD class signal decomposition methods and determined the use of the CEEMD decomposition method, extracted the sample entropy of the decomposed modal components used to characterize the sound signals, and input it into SVM to construct the sound recognition model. The authors of [6] propose the decomposition method of DTWCT to realize the decomposition and reconstruction of sound signals, extract the energy entropy ratio of its high-frequency components, which is used to characterize the sound signals, and input them into ELM to construct the sound recognition model. The authors of [7] propose the wavelet packet decomposition method to realize the decomposition of the sound signal, extract the energy ratio of its decomposition components, which is used to characterize the sound signal, and input it into the BP neural network to construct the sound recognition model.

With continuous deep learning research, sound recognition and classification have received great attention from researchers and practitioners, and sound recognition and classification have been widely applied first to environmental sound classification [8,9], noise signal classification [10], speech/music classification [11], music genre classification [12], speaker recognition [13], and indoor localization [14]. In the literature [15,16], an automatic speech recognition system and powerful sound event classification were developed using deep neural networks, and the authors of [17] utilized deep recurrent neural networks for speech recognition and sound event classification.

To further improve the accuracy of coal mine gas and coal dust explosion recognition, it is necessary to conduct in-depth research in coal mine gas and coal dust explosion sound recognition. Inspired by the research ideas of [15,16,17], we try to find a deep network suitable for actual coal mine production. Multi-layer perception (MLP) has the advantages of highly parallel processing, highly nonlinear global action, good fault tolerance, associative memory function, very strong adaptive and self-learning function, etc. Based on this, a sound identification method for gas and coal dust explosions based on MLP is proposed in this paper, which mainly includes the following aspects: (1) the real-time acquisition of sound signals in the monitoring area; (2) pre-processing of sound signals; (3) the extraction of the feature parameters of sound signals; (4) filtering less useful features by the Relief algorithm to achieve a reduction in the dimensionality of the extracted feature set; (5) establishing a machine learning recognition model; and (6) determining whether the sound is a gas and coal dust explosion by the model results. Working Principle Diagram is shown in Figure 1.

## 2. Feature Extraction

### 2.1. Sound Material

In this paper, the experimental work of underground non-gas and coal dust explosion sound data collection was conducted in the Shuangma coal mine of the Shenhua Ningxia Coal Group. The sound of gas explosions and coal dust explosions is recorded by China Coal Industry Research Institute Chongqing Co. which located Chongqing, China.

The sound acquisition equipment is an HYV-E720 recorder with 16 GB memory, and all the audio files are mono with a 48 kHz sampling rate and saved in a .wav format. The data processing and experiments were performed on a Dell server using Matlab 2020a. The server is configured with an Intel i9-9980HK CPU@2.40 GHz, 32 GB RAM, and a 64-bit operating system.

### 2.2. Sound Characteristics

Compared to a single selection of a feature, different sound signal features can better describe certain aspects of the sound signal characteristics. To achieve a more comprehensive description of the sound signal, this paper will extract the signal statistical features, time-frequency domain features, sound features, etc. To realize the effective fusion of the extracted various sound features, this paper will use a Hamming window size of 1024. The frameshift is 50% of the window size, that is, 512, and the average value of all the extracted parameters will be used as the feature parameters of the sound signal. The distribution of each parameter is specified in Table 1.

The sound signal features selected in this paper have been proven to be effective in recognition classification work in different applications. Among them, the short-time energy, zero crossing rate, and 16-dimensional time-frequency features are all time-domain features of sound signals [18]; the spectral center of mass, spectral spread, and roll-off coefficient can effectively respond to the energy distribution features of sound signals [19]; the MFCC and GFCC are two common sound features widely used in the field of speech recognition [20,21,22], and short-time Fourier coefficients can effectively respond to the frequency domain features [23].

The short-time energy is the signal energy within a given window length, which reflects the strength of the sound signal at different moments. It is calculated as
(1)$$E(i)=\frac{1}{N}\sum_{n=1}^{N}|x_{i}(n){|}^{2}.$$

where $x_{i}\left(n\right)$ is the sound signal and *N* is the signal frame length.

The spectral center of mass and spectral spread are important parameters to describe the timbre properties. The spectral center of mass is the frequency in a certain frequency range by energy weighting, which reflects the brightness of the sound signal and is calculated as
(2)$$C=\frac{\sum_{k=1}^{N}k\cdot A(k)}{\sum_{k=1}^{N}A(k)}.$$

where *k* represents the signal frequency and A(k) is the spectral amplitude within a given time window.

The spectral spread is a measure of the spectral center-of-mass distribution and is a weighted average of the spectral center-of-mass bands, which is calculated as
(3)$$S=\frac{\sum_{k=1}^{N}\left|k-C|\cdot A(k\right)}{\sum_{k=1}^{N}A(k)}.$$

where *C* is the center frequency band.

The zero crossing rate is the number of times the signal value passes through zero in a specified time, which reflects the number of times the signal passes through zero and reflects the frequency characteristics, and it is calculated as
(4)$$Zn=\sum_{m=-\infty }^{\infty }\left|\mathrm{sgn}[x(m)]-\mathrm{sgn}[x(m-1)]|w(m\right).$$

where sgn() is the sign function.
(5)$$\mathrm{sgn}[x(n)]=\left\{ \begin{array}{l} \begin{array}{cc} 1 & \end{array}x(n)\geq 0\\ \begin{array}{cc} -1 & \end{array}x(n)<0 \end{array}\right..$$

w(m) is calculated as
(6)$$w(m)=\left\{ \begin{array}{l} \begin{array}{cc} \frac{1}{2N} & 0\leq n\leq N-1 \end{array}\\ \begin{array}{cc} 0 & \mathrm{else} \end{array} \end{array}\right..$$

The roll-off coefficient is a measure of spectral skewness, which is the percentage of the signal energy in the set frequency and reflects the signal.

The spectrum roll-off is a measure of the spectral bias, which is the percentage of the signal energy below a set frequency, reflecting the distribution of the signal energy, which is calculated as
(7)$$\sum_{i}^{R}s_{i}(t)=a\cdot \sum_{i=0}^{I-1}s_{i}(t).$$

where $s_{i}\left(t\right)$ is the signal energy of the ith frame and a is the general value. *I* is the total number of signal frames. *R* is the length of the sound signal. The general range of a is [0.8, 0.9], and 0.9 is used in this paper.

Due to the limitation of space, this paper will take the gas explosion sound, coal dust explosion sound, coal mining machine working sound, roadheader working sound, and ventilator working sound as the research objects by extracting the average values of five parameters, such as the short-time energy, spectral mass center, spectral diffusion, roll-off coefficient, and zero crossing rate, and the specific distribution is shown in Figure 2.

It can be seen from Figure 2 that the mean values of the short-time energy, spectral center of mass, spectral diffusion, roll-off coefficient, and zero crossing rate of gas explosion sound and coal dust explosion sound are similar in magnitude and similar in distribution; the mean values of the short-time energy of coal mining machine working sound, roadheader working sound, and ventilator working sound are similar in magnitude, but the mean values of the remaining four characteristics are significantly different in magnitude, and the mean values of the short-time energy of the gas explosion sound and the coal dust explosion sound differed significantly from those of the other three sounds. The mean values of the roll-off coefficient and zero crossing rate differed considerably, and the mean values of the spectral center of mass and spectral spread did not differ much.

The time-domain features selected in this paper include the mean value, root mean square, root square amplitude, absolute mean, skewness, cliffness, variance, maximum value, minimum value, peak-to-peak value, waveform index, peak index, pulse index, margin index, skewness index, cliffness index, etc. There are a total of 16 parameters, which can describe the time-domain features of sound signals more comprehensively. The average values of the time-domain characteristics of gas explosion sound, coal dust explosion sound, coal mining machine working sound, roadheader working sound, and ventilator working sound are also extracted, and their specific distributions are shown in Figure 3.

As can be seen from Figure 3, the average values of the time-domain features of gas explosion sound and coal dust explosion sound are similar in size except for the small differences in the average values of three parameters, namely the pulse index, margin index, and skewness index. The average values of the 16 parameters of the time-domain features of the coal mining machine working sound, coal mining machine working sound, and ventilator working sound have small differences and are highly similar. The time-domain characteristics of the gas explosion sound and coal dust explosion sound differ significantly from those of the other three sounds.

In addition, in this paper, in order to characterize the sound signal in more detail, the frequency-domain features that are widely used in the field of sound recognition are also selected, including MFCC, GFCC, and short-time Fourier coefficients, and the three frequency-domain features have been widely used in the fields of speech recognition, speech emotion recognition, and security monitoring.

MFCC mainly adopts the new metric *Mel* value, which is closer to the hearing mechanism of the human ear than the frequency, and the conversion relationship between the *Mel* value and the frequency value is shown in the following equation:(8)$$f_{Mel}=2595\cdot \log(1+\frac{f}{700}).$$

where *f* is the frequency.

The process of *Mel* inverse spectrum coefficient extraction is: (1) adding a window and splitting frame: first, the sound signal is split into frames, which can increase the continuity between each frame of sound, and each frame of the signal after splitting the frame is multiplied by the window function for filtering; (2) the windowed signal is passed through the fast Fourier transform to obtain the energy distribution of the signal on the spectrum; (3) the discrete power spectrum is calculated. The log energy is obtained by filtering through a set of Meier filters; and (4) the log energy calculated in step (3) is discrete cosine transformed to obtain the MFCC coefficients.

The feature extraction of GFCC is based on a more comprehensive model of an equivalent rectangular bandwidth scale and a set of Gammatone filters, which simulates the process of the human auditory system for sound signals, and its coefficient extraction process is as follows: steps (1)~(2) are the same MFCC coefficient extraction process; (3) pass the energy spectrum through a set of Gammatone filters; (4) calculate the short-time log energy; and (5) process the log energy by discrete cosine transform to obtain the GFCC coefficients.

The short-time Fourier coefficients realize the connection between the time and frequency domains of the signal, and the process is to multiply a time-limited window function *h(t)* before the Fourier transform of the signal and assume that the non-stationary signal is stationary in the short interval of the analysis window. The short-time Fourier transform of the signal *x(τ)* is defined as
(9)$$STFT(t,f)=\int_{-\infty }^{\infty }x(\tau )h(\tau -t)e^{-j2\pi f\tau }d\tau .$$

where STFT(t,f) is the spectrum at time *t* and h(τ−t) is the analysis window.

According to the previous theoretical analysis, the average values of the MFCC eigenvalues, GFCC eigenvalues, and short-time Fourier coefficients of gas explosion sound, coal dust explosion sound, coal mining machine working sound, coal mining machine working sound, and ventilator working sound are also extracted, and their specific distributions are shown in Figure 4, Figure 5 and Figure 6.

As can be seen from Figure 4, the mean values of the MFCC eigenvalues of gas explosion sound and coal dust explosion sound are similar in size and have a similar distribution. The mean values of the MFCC eigenvalues of the working sound of the coal mining machine, roadheader, and ventilator have a significant difference in size in the first 6 dimensions of the MFCC eigenvalues, and the remaining 6 dimensions do not differ greatly in size. The mean values of gas explosion sound, coal dust explosion and working sound of coal mining machine, roadheader, and ventilator have large differences in the 1st, 4th, 6th, 7th, 9th, and 12th dimensions, and the differences in the remaining 6 dimensions are not significant.

As can be seen from Figure 5, the mean values of GFCC eigenvalues of gas explosion sound and coal dust explosion sound are similar in size and have a similar distribution; the mean values of GFCC eigenvalues of working sound of coal mining machine and roadheader are similar in size in the first six dimensions of the mean GFCC eigenvalues, with little difference in size in the remaining six dimensions; the mean values of the GFCC eigenvalues of the working sound of the ventilator and roadheader in the last six dimensions are similar in value, and there is little difference in the value in the remaining six dimensions. The mean values of the GFCC eigenvalues of gas explosion sound, coal dust explosion sound, and working sound of the coal mining, roadheader, and ventilator have some similarities; there are small differences in the magnitudes of the values.

From Figure 6, it can be seen that the mean values of the short-time Fourier coefficients of gas explosion sound and coal dust explosion sound are similar in size, and the distribution is more similar.The mean values of the short-time Fourier coefficients in the 1st, 2nd, 4th, 5th, and 6th dimensions of the working sound of the coal mining machine have a large difference in size. The mean values of the short-time Fourier coefficients in the 1st, 2nd, 4th, 5th, and 6th dimensions of the working sound of the roadheader have a large difference in size, and the mean values of the short-time Fourier coefficients in the first six dimensions of the working sound of the ventilator have a large difference in size. The remaining 19 dimensions of the short-time Fourier coefficient average values are similar, and the similarity is high.

Through the above analysis, it can be seen that most of the 70-dimensional feature parameters selected in this paper have large differences, and there are still a small number of feature parameters with too high similarity, which will greatly hinder the establishment of the recognition model and affect the accuracy and precision of the recognition model. Therefore, it is necessary to achieve the establishment of the optimal recognition model by selecting the best features as the characterization sound.

### 2.3. Optimal Feature Extraction

In this paper, a total of 70 features of commonly used sound signals are selected. To select the features that are more compatible with the sound signals, this paper will realize the selection of features using the Relief algorithm, whose main idea is to assign weights to each feature and then sort them, and the weight of each feature is calculated based on the homogeneity of neighboring features. The specific process is as follows:(1)Loading the feature dataset into the code for pre-processing, removing the duplicate items, and recording the features of the sound signal, the category of the sound signal, and its parameters, respectively;(2)Calling the features of the sound signal, the category of the sound signal, and their parameters by the Relief algorithm, and the output of this function is an idx table containing the features sorted in descending order of importance and a table containing their weights;(3)The data are shuffled using the pseudo-random number generation function, randperm. Before each call to the function, the command rng (0) is invoked to ensure the same initialization of the random process and obtain the same result in each execution of the program;(4)Based on the predetermined judging index, suitable parameters are selected as features to characterize the sound signal.

To objectively evaluate the best feature extraction method proposed in this paper, a fivefold cross-validation scheme is used, i.e., the training samples are divided into five parts, i.e., the data are equally divided into five equal parts, and one of them is taken for testing and the others for training each time. After training the recognition model and calculating the cross-entropy based on the prediction results of the model in the test and validation set entropy, we find the feature parameter corresponding to the smallest cross-entropy, which is the best point of the number of feature parameters; according to the distribution of the cross-entropy around the best point, we select the number of feature parameters with similar cross-entropy and input them into the training model together and select the best number of features by the recognition according to the distribution of the cross-entropy around the best point. The number of features with similar cross-entropy is selected and input to the training model, and the optimal number of features is selected through the recognition result to achieve the optimization of the model.

## 3. Recognition Model Establishment

### 3.1. Sound Data

The training samples include 5 groups of each sound, resulting in 80 groups in total; the test samples include 100 groups of each sound, resulting in 1600 groups in total.

### 3.2. Feature Parameter Determination Test

In this paper, the determination of the optimal number of features is achieved using MLP [24]; therefore, this paper needs to select the number of neurons, a training function, and a performance function of each hidden layer, and the specific parameters of the model function are specified as follows:(1)After experimental verification, the ideal number of neurons for the first hidden layer and the second hidden layer are 20 and 80, respectively.(2)Training function: The network training function selected in this paper is Trainrp, which means the backpropagation method. Therefore, the weight update of the network training is performed by minimizing the cost function.(3)Performance function: This paper uses cross-entropy as the quality evaluation index of the network performance [25]. In the classification model, identification as a certain class belongs with a probability of 1, and for other classes, it belongs with a probability of 0. Each model estimates the probability that a record belongs to a certain class. The cross-entropy is the difference between two distributions. It is minimized in the same way as the likelihood function is maximized.

In this paper, according to the steps in Section 2.3, to select the best feature parameters in the Relief algorithm output idx table to take the first 5, 10, 15, ..., 65, and 70 feature parameters into the MLP and calculate the cross-entropy distribution obtained for each group of feature parameters, the input model training is shown in Figure 7. As can be seen from Figure 7, the light blue marked point is the point with the smallest global cross-entropy value, corresponding to the number of feature values of 60 when the value of cross-entropy is 0.01184; however, the value of cross-entropy corresponding to the feature parameters in the red box line is less different from that when the number of feature values is 60. Therefore, in this paper, the recognition results corresponding to the number of feature values in the red box will be discussed and verified to select the optimal number of feature values.

To select the best feature parameters for the model training, this paper also discusses the final recognition results of the recognition model after the corresponding feature parameters are input into the MLP to build the voice recognition model when the number of feature parameters is 10, 20, 25, 35, 45, and 60, and the specific results are shown in Figure 8. As can be seen from Figure 8, when the number of feature values is 60, the recognition result of the trained recognition model is the worst among several comparison models; therefore, the final recognition result of the recognition model is not directly related to the number of feature values. The recognition result of the sound recognition model proposed in this paper is better in gas and coal dust explosion sound, coal mining machine working sound, scraper working sound, and reloader working sound. The recognition results of the emulsion pump working sound, digging machine working sound, anchor machine working sound, wind pick working sound, mobile substation working sound, and rubber wheel car driving sound are poor; other sounds can be recognized by the recognition model after selecting suitable feature values.

After analyzing the cross-entropy distribution and the overall recognition effect of the sound recognition model, it can be seen that the best sound recognition model can be trained when the number of feature values is taken as the first 35 feature values and input to MLP.

### 3.3. MLP Model Building

MLP is a forward-structured artificial neural network that maps a set of input vectors to a set of output vectors, and the MLP can be viewed as a directed graph consisting of multiple node layers, each of which is fully connected to the next. In addition to the input nodes, each node is a neuron with a nonlinear activation function. A supervised learning approach using a BP (backpropagation algorithm) is used to train the MLP, which is a generalization of the perceptron and overcomes the weakness that the perceptron cannot recognize linearly indistinguishable data.

In the training phase of MLP, the first 35-dimensional feature values determined in Section 2.1 are input into the MLP to initialize the weight matrix and bias coefficients. There is a total of p neurons in layer m−1 of the network, and then the output value of the jth neuron in the mth layer is
(10)$$\alpha_{j}=g(\sum_{i=1}^{p}w_{i}^{m}\alpha_{i}^{m-1}+b_{i}^{m}).$$

where $W_{i}^{m}$ is the weight of the ith neuron in the mth layer, $\alpha_{i}^{m-1}$ is the output value of the ith neuron in the (*m*−1)th layer, $b_{i}^{m}$ is the bias vector of the jth neuron in the mth layer, and *g*(·) is the nonlinear activation function.

The error is calculated using the output value, and the sample is then used to update the weights using backpropagation until the output error is lower than the preset value. The output error is calculated as follows:(11)$$LOSS=\frac{1}{2}{\sum_{p=1}^{M}\left(y_{p}-\hat{y_{p}}\right)}^{2}.$$

where $y_{p}$ is the true label and $\hat{y_{p}}$ is the predicted value.

Finally, the test sample set is fed into the completed training MLP model to obtain the final sound classification results.

## 4. Results

To quantitatively evaluate the proposed sound recognition method, the recognition rate, accuracy rate, and recall rate are used as the evaluation criteria of the proposed method.

The accuracy rate refers to the probability that all the samples with positive predictions are positive samples, which is calculated using the following formula:(12)$$precision=\frac{TP}{TP+FP}.$$

where *TP* represents the number of positive predicted samples and the actual number of positive samples, and *FP* represents the number of positive predicted samples and the actual number of negative samples.

Recall refers to the probability of a positive sample being predicted out of an actual positive sample and is calculated as
(13)$$recall=\frac{TP}{TP+FN}.$$

where *FN* is the number of predicted negative and actual positive samples.

### 4.1. Model Runtime

In practical engineering applications, the accurate and efficient identification of coal mine gas and coal dust explosions is crucial for coal mine safety production and safety rescue work. This experiment was performed on a DELL server with an Intel i9-9980HK CPU@2.40GHz, 32Gb memory, and a 64-bit operating system using MATLAB2020a. To avoid errors, the results of each test were averaged by performing 10 repetitions of the experiment, and the running time of each part of the proposed method in this paper is shown in Table 2. As can be seen from Table 2, the best feature extraction in this paper takes the most time, which is about 15.52 s. The feature extraction, MLP training, and recognition take 0.05 s, 0.83 s, and 0.62 s, respectively. Considering that the best feature extraction can be conducted again during non-working hours, it does not affect the normal working time. This method can complete the feature extraction, model training, and recognition process in less time on a less configured hardware platform, which has high computational efficiency and meets the requirement of the real-time safety monitoring and alarm system in underground coal mines.

### 4.2. Comparison of Results

To evaluate the proposed sound recognition method for coal mine gas and coal dust explosions, this group of experiments also took five groups of each sound (80 groups in total) to form the training sample set and 100 groups of each sound (1600 groups in total) to form the test sample set. The training sample set is trained by feature extraction, best feature value extraction, and the MLP model to obtain the recognition model, and the test sample set is input to the recognition model, and the final recognition results are shown in Figure 9.

As can be seen from Figure 9, the recognition rate and recall rate of the algorithm proposed in this paper are higher than 90% for all the sounds, except the sound of the working sound of the roadheader, wind pick working, and mobile substation working, and the accuracy rate of all the sounds, except the working sound of the emulsion pump, anchor machine, and mobile substation working, is higher than 90%. The average recognition rate of the algorithm proposed in this paper reaches 95%. The average recall rate is 95%, and the average accuracy rate is 95.8%.

To verify the advantages of the coal mine gas and coal dust explosion sound recognition method proposed in this paper, the recognition results of the algorithm in this paper are compared with the coal mine gas and coal dust explosion sound recognition results proposed in the literature [5,6,7], and the specific comparison results are shown in Table 3. As can be seen from Table 3, the algorithm proposed in this paper has a recognition rate of 95%, which is 10% and 2% higher than that in [5,6], respectively and is the same as that in [7]; the recall rate in [6] is up to 100%, which is 5% higher than that of the algorithm proposed in this paper and 16.7% and 25% higher than that in [5,7], respectively. The algorithm proposed in this paper has an accuracy rate of 95.8%, which is higher than that in [5,6], 24.4% and 14.7% higher than in [5] and 4.2% lower than in [7], respectively. Comprehensive analysis shows that the recognition effect of the algorithm proposed in this paper is significantly better than the compared methods, and the algorithm proposed in this paper can effectively distinguish each kind of voice involved in the experiment.

## 5. Conclusions

(1)In this paper, a sound identification method for gas and coal dust explosions based on MLP was proposed. The distributions of the short-time energy, zero crossing rate, spectral center-of-mass parameters, spectral spread, roll-off coefficient, 16-dimensional time-frequency features, MFCC, GFCC, and the average of the short-time Fourier coefficients of 16 sound signals, including coal mine gas and coal dust explosion sounds collected in the field, are analyzed, which can effectively distinguish coal mine gas and coal dust explosion sounds from non-coal mine gas and coal dust explosion sounds.(2)The best feature extraction model is established, and the influence of different numbers of feature value parameters on the model training situation and recognition results is analyzed. With the cross-entropy and model recognition rate as the evaluation objects, the best feature parameters can be selected to avoid the influence of feature parameters with poor discrimination on the model training, and the compatibility and portability of this method can be effectively solved.(3)The experimental results show that the proposed algorithm can effectively distinguish each kind of sound signal participating in the experiment, and the average recognition rate reaches 95%. In addition, the method proposed in this paper can be used not only for the intelligent recognition of coal mine gas and coal dust explosions but also for monitoring abnormal sounds in underground coal mines; by modifying the training data set, it can also be used for monitoring abnormal sounds in other large public places, such as tunnels and subways.

## Figures and Tables

**Figure 1 entropy-25-01184-f001:**
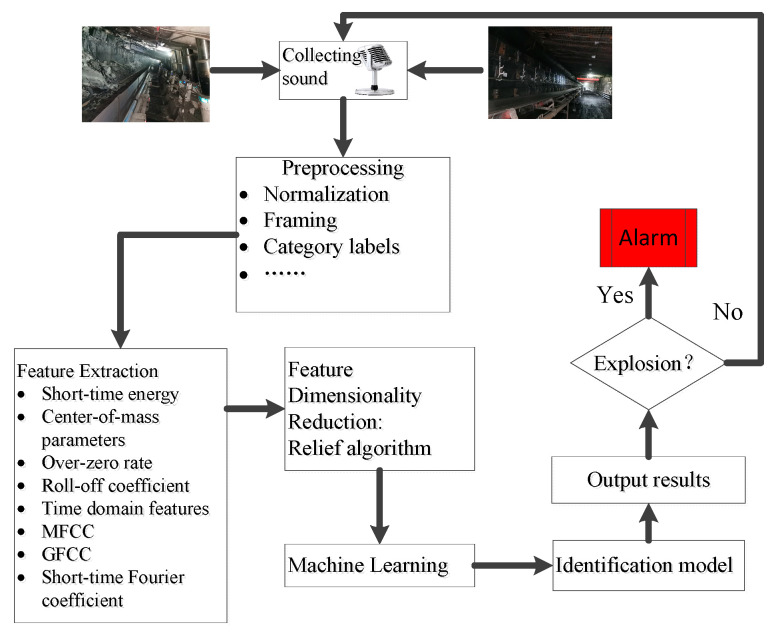
Working Principle Diagram.

**Figure 2 entropy-25-01184-f002:**
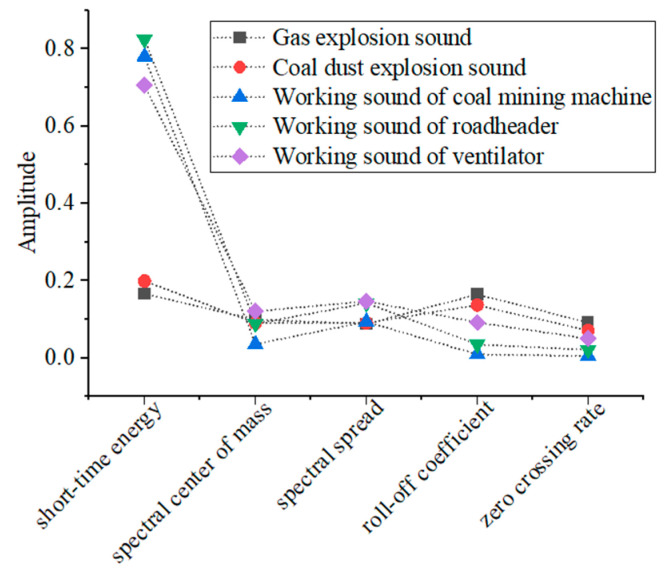
The mean value distribution of characteristics.

**Figure 3 entropy-25-01184-f003:**
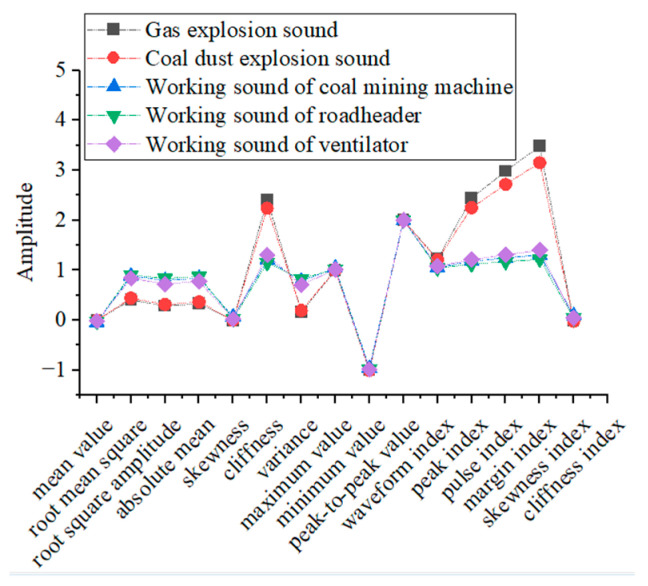
The mean value distribution of time-domain features.

**Figure 4 entropy-25-01184-f004:**
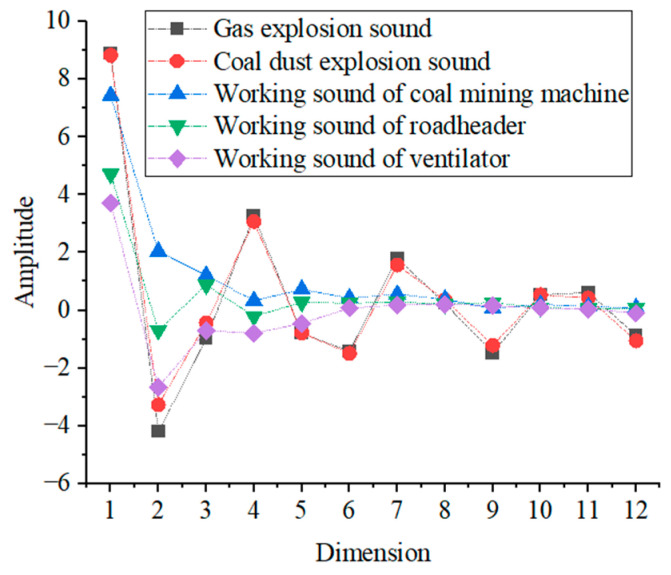
The mean value distribution of MFCC.

**Figure 5 entropy-25-01184-f005:**
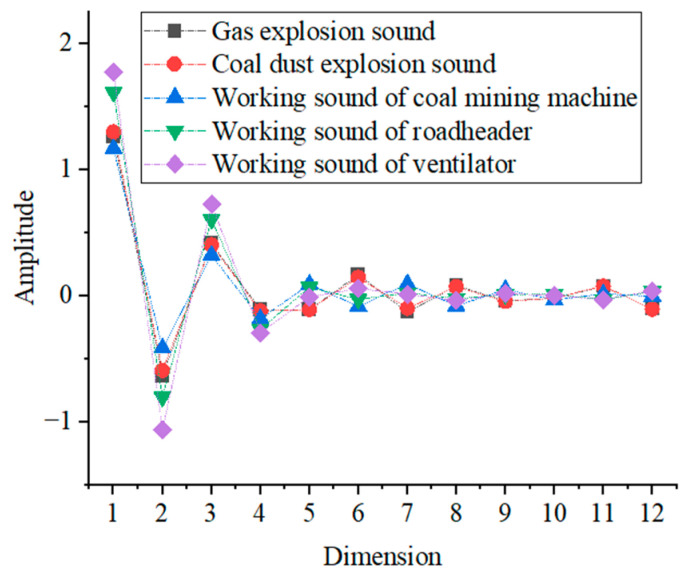
The mean value distribution of GFCC.

**Figure 6 entropy-25-01184-f006:**
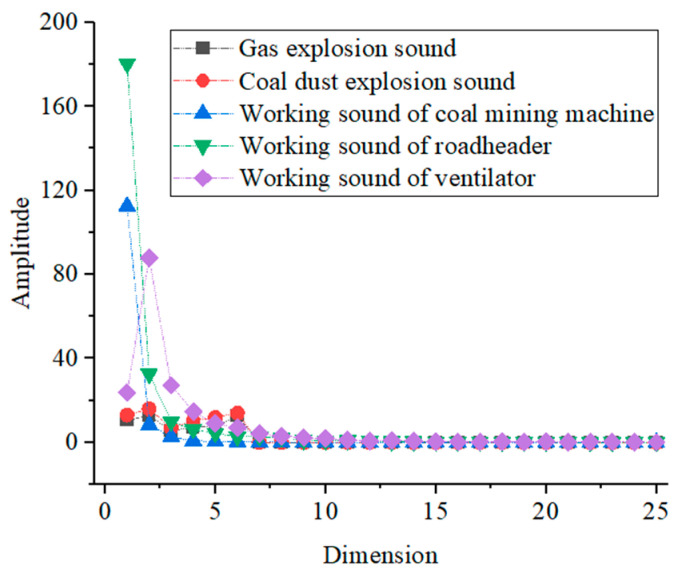
The mean value distribution of short-time Fourier coefficients.

**Figure 7 entropy-25-01184-f007:**
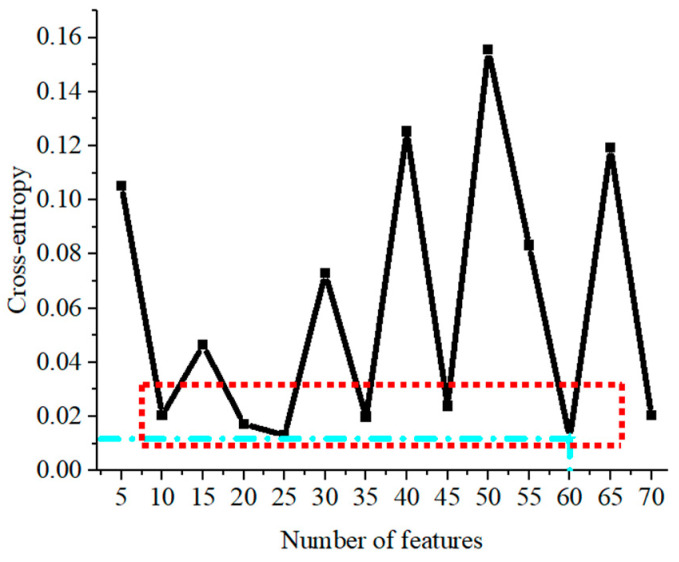
The distribution of cross-entropy.

**Figure 8 entropy-25-01184-f008:**
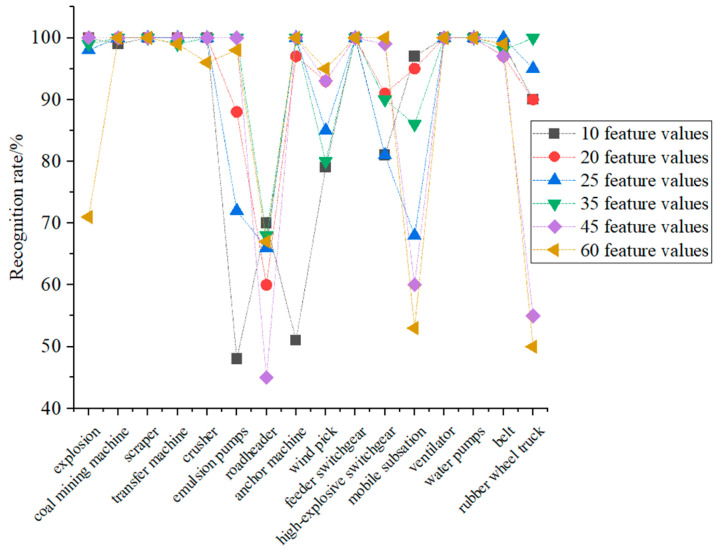
Recognition model recognition results.

**Figure 9 entropy-25-01184-f009:**
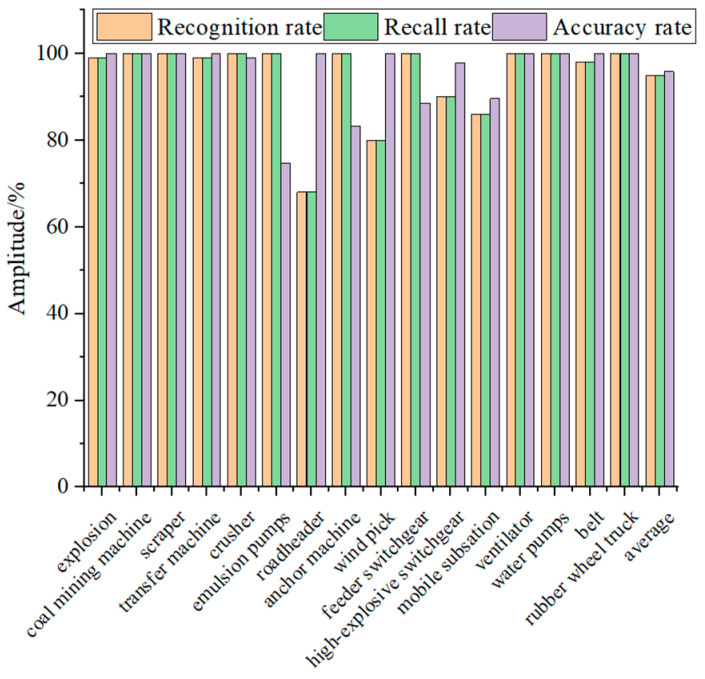
Identification results.

**Table 1 entropy-25-01184-t001:** Extracted features.

Feature Name	Dimension
Short-time energy	1
Center-of-mass parameter	2
Zero crossing rate	1
Roll-off coefficient	1
Time-domain feature	16
MFCC	12
GFCC	12
Short-time Fourier coefficient	25
Total	70

**Table 2 entropy-25-01184-t002:** Time consumption.

Model	Time Consumption/s
Feature extraction (single sample)	0.05
Optimal feature extraction (80 training samples)	15.52
MLP training (80 training samples)	0.83
MLP recognition (1600 training samples)	0.62

**Table 3 entropy-25-01184-t003:** Recognition results of classification models.

Model	Recognition Rate/%	Recall Rate/%	Accuracy Rate/%
Methods in this paper	95	95	95.8
Literature 5	85	83.3	71.4
Literature 6	93	100	81.1
Literature 7	95	75	100

## Data Availability

Data sharing not applicable.

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
