# Peer review of "Sound Identification Method for Gas and Coal Dust Explosions Based on MLP"

_entropy, 2023, doi:10.3390/e25081184_

Round 1
Reviewer 1 Report
In the manuscript, the author proposes an MLP-based coal mine gas and dust explosion recognition method, combined with the best feature extraction model to analyze the cross-entropy distribution of the MLP model after training different feature values. The manuscript has several fundamental weaknesses that have been pointed out as follows. The more notable ones are associated with the clarity of presentation, technical detail, and comparative evaluations. I have provided several well-founded comments that describe these points in detail. All things considered, my recommendation is to reject the manuscript in its current form. The overall quality of the manuscript is poor, and the review opinions are as follows.
1. Abstract The first sentence proposes an MLP-based coal mine gas and dust explosion recognition method. Is this method proposed by itself? Or is it proposed by others? If it was proposed by others, where is the innovation of this paper?
2. Abstract The second sentence says that the first 35 -dimensional special value is determined as a feature vector to characterize the sound signal. Is it reflected in the following manuscript?
3. Abstract The third sentence says that this method is to complete the feature extraction, model training, and identification process in less time, with high calculation efficiency, but no comparison in the following manuscript. How can reflect less time? efficient?
4. Abstract does not mention the sound feature extraction of the sound characteristics in the use of the RELIEF algorithm in the following?
5. All formulas need to be added with punctuation.
6. The curves in many pictures in the manuscript overlap, and it is difficult to distinguish the curve changes of different colors.
7. 2.2 chapters The spectrum quality heart, spectrum diffusion and excessive zero rate of the gas explosion and coal dust explosion in Figure 2 are significantly different from the average value of other three sounds? There is no significant difference in the picture。
8. 2.3 chapters The characteristics of the sound signal, the category, and the parameters of the sound signal are called through the RELIEF algorithm, and the sound feature extraction algorithm has a square variance method and the correlation coefficient method. Why use the RELIEF algorithm? What is the difference?
9. In the algorithm section, the author said that he proposed the use of BP genetic algorithms to train MLP. Where is the improved part? What are the innovation points of the entire algorithm?
10. The conclusion partly proposes to effectively distinguish the explosion of coal mines gas and dust explosion and non -coal mine gas and dust explosion. There is no difference sound of coal mines and the sound of non-coal mines.
Moderate editing of English language required。
Author Response
Point 1. Abstract The first sentence proposes an MLP-based coal mine gas and dust explosion recognition method. Is this method proposed by itself? Or is it proposed by others? If it was proposed by others, where is the innovation of this paper?
Response: Sound identification method for gas and coal dust explosion based on MLP was proposed in this paper, which aims to provide an accurate alarm for coal mine gas and coal dust explosion and provide a new method for coal mine safety production monitoring. The methodology was proposed after reading a large amount of literatures and it has not been used in the field of sound recognition of coal mine gas and coal dust explosions. its main innovations include : (1) The time-domain features and frequency-domain features of the sound of gas and coal dust explosions and other sounds of coal mines, totaling 70 dimensions, were analyzed, and the RELIEF algorithm was proposed to achieve the selection of the best features. (2) The sound recognition model of coal mine gas and coal dust explosion is constructed, the recognition experimental results are analyzed, and it is proved that the method can meet the intelligent recognition requirements of coal mine gas and coal dust explosion. (3) Combining sound recognition with image recognition to visualize the characteristics of sound signals in underground coal mines for the purpose of coal mine gas and coal dust explosion identification.
Point 2. Abstract The second sentence says that the first 35 -dimensional special value is determined as a feature vector to characterize the sound signal. Is it reflected in the following manuscript?
Response: In 3.2 (L293-L341) discussed in detail how the selection of the best feature parameters can be achieved by using the Relief algorithm, and also why this paper chooses the first 35 -dimensional special value is determined as a feature vector to characterize the sound signal.
Point 3. Abstract The third sentence says that this method is to complete the feature extraction, model training, and identification process in less time, with high calculation efficiency, but no comparison in the following manuscript. How can reflect less time? efficient?
Response: In 4.1 (L393-394) time consumption shows feature extraction (single sample) took 0.05s , optimal feature extraction (80 training samples) took 15.52s , the average time is 0.194s , MLP training (80 training samples) took 0.83s , the average time is 0.1s , MLP recognition (1600 training samples) took 0.62s , the average time is 0.0004s . From the above data, the proposed algorithm has high computational efficiency and meets the requirement of real-time coal mine safety monitoring and alarm system.
Point 4. Abstract does not mention the sound feature extraction of the sound characteristics in the use of the RELIEF algorithm in the following?
Response: According to the expert's opinion, The original text was added as follows(The added parts are highlighted in yellow in the paper L15):
“the best feature extraction model of Relief algorithm was established”
Point 5. All formulas need to be added with punctuation.
Response: According to the expert's opinion, all formulas added with punctuation.
Point 6. The curves in many pictures in the manuscript overlap, and it is difficult to distinguish the curve changes of different colors.
Response: Pictures in the manuscript overlap indicate that different eigenvalues of different sound signals have similar values, suggesting the need to choose appropriate eigenvalues to characterize the sound signal, proving the importance of the best feature extraction model of Relief algorithm was established
Point 7. 2.2 chapters The spectrum quality heart, spectrum diffusion and excessive zero rate of the gas explosion and coal dust explosion in Figure 2 are significantly different from the average value of other three sounds? There is no significant difference in the picture.
Response: According to the expert's opinion, The original text was added as follows (The added parts are highlighted in yellow in the paper L157):
“and the mean values of the short-time energy of the gas explosion sound and the coal dust explosion sound differed significantly from those of the other three sounds, the mean values of the spectral roll-off and the over-zero rate differed considerably, and the mean values of the spectral center of mass and the spectral diffusion did not differ much.”
Point 8. 2.3 chapters The characteristics of the sound signal, the category, and the parameters of the sound signal are called through the RELIEF algorithm, and the sound feature extraction algorithm has a square variance method and the correlation coefficient method. Why use the RELIEF algorithm? What is the difference?
Response: Reasons for use the Relief algorithm: (1) construct the optimal feature selection model construction, complete the independent selection of sound features, (2) Verify the best feature selection by using cross entropy and recognition rate as inspection indexes, and complete the optimal selection of sound features.
Point 9. In the algorithm section, the author said that he proposed the use of BP genetic algorithms to train MLP. Where is the improved part? What are the innovation points of the entire algorithm?
Response: Supervised learning component using BP genetic algorithms, the innovation points of the entire algorithm: (1) Propose Relief algorithm to construct the optimal feature selection model construction, complete the independent selection of sound features, (2) Verify the best feature selection by using cross entropy and recognition rate as inspection indexes, and complete the optimal selection of sound features.
Point 10. The conclusion partly proposes to effectively distinguish the explosion of coal mines gas and dust explosion and non -coal mine gas and dust explosion. There is no difference sound of coal mines and the sound of non-coal mines.
Response: Gas explosions, coal dust explosions, and gas and coal dust explosions (hereafter re-ferred to as gas and coal dust explosion) are one of serious accidents in coal mines, which will cause casualties and economic losses once they occur. The existing means of monitoring coal mine gas and coal dust are mainly realized by gas and temperature sensors, which are outdated monitoring means and have problems such as slow reporting, false alarm rate, and high leakage rate. The manuscript aims to provide an accurate alarm for coal mine gas and coal dust explosion and provide a new method for coal mine safety production monitoring.

Reviewer 2 Report
This paper proposes a MLP-based sound identification method for gas and dust explosion detection in coal mines. The detection accuracy of actual coal mine gas and coal dust explosions is verified through the investigation of the optimal feature extraction model for sound identification and the dimensionality of the features.
The occurrence of gas and dust explosion accidents in coal mines is likely to have serious consequences, and detection and warning technologies to prevent explosions from occurring are the first priority, but they are required to prevent the spread of damage.
Is it technically possible to estimate the distance of the explosion source from the observation location based on frequency-domain and time-domain feature analysis?
The following are individual comments.
L145 equation 7; variable R (roll off?) and Large-I should be explained.
L160; ... spectral center of mass, spectral diffusion, ... are significantly different from those of other three sounds.; Spectral center of mass, spectral diffusion ('spread' in the figure) appear to be within the range of variation of the other three amplitude values, but how do they differ?
L181 Fig.3; "mean value" on the in x-axis; Is "mean value" a description of the entire X axis rather than tick labels on the axis?
L221; the mean values of MFCC eigenvalues of working sound of coal mining machine, roadheader, and ventilator are similar in size in the first 6 dimensions..., and the remaining 6 dimensions do not differ greatly in size; I am not sure what difference you are trying to explain with the phrases "similar in size" and "not differ greatly in size."
L224; the mean values of gas explosion sound and coal dust explosion sound are similar in size to the mean values of coal mining machine working sound, "coal dust explosion sound" and working sound of ventilator.; What is the meaning of "coal dust explosion sound" in the second half?
L235; are similar in size in the first 6 dimension... little difference in size in the remaining 6 dimensions; L221; As in the case of L221, the difference between "similar in size" and "little difference" is unclear.
L237; are similar in value, and there is little difference in value size; Similar to the points made in L221.
L240; have some similarity, but there are large differences in the magnitude of the values.; Similar to the points made in L221. "Similarity" can be taken to mean the distribution trend of amplitude with respect to dimension, but it needs to be well explained.
L322 Fig. 7; It appears that cross-entropy begins to oscillate around feature 25-30, does this mean that it is starting to over-fitting in this area?
L361 Equation 11; Wouldn't there be a hat on either y on the right side?
Where does the p take as a subscript?
L382; Inter; Intel?
Author Response
Point 1. L145 equation 7; variable R (roll off?) and Large-I should be explained.
Response: According to the expert's opinion, The original text was added as follows (The added parts are highlighted in yellow in the paper L145):
“I is the total number of signal frames”
Point 2.L 57; L160; ... spectral center of mass, spectral diffusion, ... are significantly different from those of other three sounds.; Spectral center of mass, spectral diffusion ('spread' in the figure) appear to be within the range of variation of the other three amplitude values, but how do they differ?
Response: According to the expert's opinion, The original text was added as follows (The added parts are highlighted in yellow in the paper L157):
“and the mean values of the short-time energy of the gas explosion sound and the coal dust explosion sound differed significantly from those of the other three sounds, the mean values of the spectral roll-off and the over-zero rate differed considerably, and the mean values of the spectral center of mass and the spectral diffusion did not differ much.”
Point 3. L181 Fig.3; "mean value" on the in x-axis; Is "mean value" a description of the entire X axis rather than tick labels on the axis?
Response: Fig3 "mean value" on the y-axis, x-axis means parameterization.
Point 4. L221; the mean values of MFCC eigenvalues of working sound of coal mining machine, roadheader, and ventilator are similar in size in the first 6 dimensions..., and the remaining 6 dimensions do not differ greatly in size; I am not sure what difference you are trying to explain with the phrases "similar in size" and "not differ greatly in size."
Response: The difference here refers to the numerical magnitude of the eigenvalues.
Point 5. L224; the mean values of gas explosion sound and coal dust explosion sound are similar in size to the mean values of coal mining machine working sound, "coal dust explosion sound" and working sound of ventilator.; What is the meaning of "coal dust explosion sound" in the second half?
Response: According to the expert's opinion, The original text was added as follows (The added parts are highlighted in yellow in the paper L224):
“The mean values of gas explosion sound , coal dust explosion and working sound of coal mining machine, roadheader, and ventilator have large differences in the 1st, 4th, 6th, 7th, 9th, and 12th dimension, and the differences in the remaining 6 dimensions are not significant.”
Point 6. L235; are similar in size in the first 6 dimension... little difference in size in the remaining 6 dimensions; L221; As in the case of L221, the difference between "similar in size" and "little difference" is unclear.
Response: The difference here refers to the numerical magnitude of the eigenvalues.
Point 7. L237; are similar in value, and there is little difference in value size; Similar to the points made in L221.
Response: The difference here refers to the numerical magnitude of the eigenvalues.
According to the expert's opinion, The original text was added as follows (The added parts are highlighted in yellow in the paper L237):
“there have small differences in the magnitude of the values.”
Point 8. L240; have some similarity, but there are large differences in the magnitude of the values.; Similar to the points made in L221. "Similarity" can be taken to mean the distribution trend of amplitude with respect to dimension, but it needs to be well explained.
Response: According to the expert's opinion, The original text was added as follows (The added parts are highlighted in yellow in the paper L243):
“The mean values of short-time Fourier coefficients in the 1st, 2nd, 4th, 5th, and 6th dimension of the sound of coal mining machine work have a large difference in size, the mean values of short-time Fourier coefficients in the 1st, 2nd, 4th, 5th, and 6th dimension of the sound of roadheader machine work have a large difference in size, and the mean values of short-time Fourier coefficients in the first 6 dimensions of the sound of the ventilator work have a large difference in size. The remaining 19 dimensions of the short-time Fourier coefficient average values are similar, and the similarity is high.”
Point 9. L322 Fig. 7; It appears that cross-entropy begins to oscillate around feature 25-30, does this mean that it is starting to over-fitting in this area?
Response: To prevent the situation of over-fitting mentioned by the experts, two parameter metrics are chosen for optimal feature selection, respectively dimensional cross entropy and recognition rate. Combined with the final model recognition experiments, it can be seen that there is no occurrence of the situation of over-fitting.
Point 10. L361 Equation 11; Wouldn't there be a hat on either y on the right side? Where does the p take as a subscript?
Response: Confirm that the formula is correct.
Point 11. L382; Inter; Intel?
Response: According to the expert's opinion, The word “Inter” was replace with “Intel” (The added parts are highlighted in yellow in the paper L382)

Round 2
Reviewer 1 Report
1. In section 2.1, "-The sound of gas explosions and coal dust explosions is recorded by China Coal Industry Research Institute Chongqing Co." Before the sentence, "-"Should it be removed?"
2. Does A(K) in Equations 2 and 3 mean the same thing? Why is the same font different? What do the different words mean?
3. What does R mean in Equation 7? Should be explained.
4. Is
in Equation 9 the x(t) mentioned above? Should be explained.
5. Is Equation 11 correct? There is no
explained below in the formula.
6. Does Equation 10 correspond to the interpretation of L351 and 352?
7. Figure 4: the mean values of MFCC eigenvalues of working sound of coal mining machine, roadheader, and ventilator are similar in size in the first 6 dimensions of MFCC eigenvalues? The figure shows that the size difference of the first 6 dimensions is obvious.
8. The curves in the pictures in the manuscript overlap each other, making it difficult to distinguish the changes of different curves. There was no good response.
9. In the algorithm section, it is recommended to use the BP genetic algorithm to train MLP. Where are the improvements? There was no good response and further explanation was needed from the author.
10. The conclusion part proposes that it can effectively distinguish between coal mine gas explosion sound and non-coal mine gas explosion sound. But there is no distinction in the text between the sound of a coal mine explosion and the sound of a non-coal mine explosion? There was no good response and further explanation was needed from the author.
Author Response
Response to Reviewer 1 Comments :
Point 1. In section 2.1, "-The sound of gas explosions and coal dust explosions is recorded by China Coal Industry Research Institute Chongqing Co." Before the sentence, "-"Should it be removed?"
Response: According to the expert's opinion, the original text was removed “-”.( The modified parts are highlighted in yellow in the paper L86)
Point 2. Does A(K) in Equations 2 and 3 mean the same thing? Why is the same font different? What do the different words mean?
Response: According to the expert's opinion, the original text was modified (The modified parts are highlighted in yellow in the paper L118, L123), and the original text was added as follows(The modified parts are highlighted in yellow in the paper L124)
“Where C is the center frequency band.”
Point 3. What does R mean in Equation 7? Should be explained.
Response: According to the expert's opinion, The original text was added as follows(The modified parts are highlighted in yellow in the paper L140)
“R is the length of the sound signal.”
Point 4. Is in Equation 9 the x(t) mentioned above? Should be explained.
Response: According to the expert's opinion, The original text was modified as follows(The modified parts are highlighted in yellow in the paper L206)
“The short-time Fourier transform of the signal x(t) is defined as”
Point 5. Is Equation 11 correct? There is no explained below in the formula.
Response: According to the expert's opinion, The original text was modified as follows(The modified parts are highlighted in yellow in the paper L358-L359)
“ (11)
Where is the true label and is the predicted value.”
Point 6. Does Equation 10 correspond to the interpretation of L351 and 352?
Response: Equation 10 correspond to the interpretation of L351 and 352. The original text was modified as follows(The modified parts are highlighted in yellow in the paper L352)
“where, is the weight of the ith neuron in the mth layer, is the output value of the ith neuron in the (m-1)th layer, is the bias vector of the jth neuron in the mth layer, and g(×) is the nonlinear activation function.”
Point 7. Figure 4: the mean values of MFCC eigenvalues of working sound of coal mining machine, roadheader, and ventilator are similar in size in the first 6 dimensions of MFCC eigenvalues? The figure shows that the size difference of the first 6 dimensions is obvious.
Response: According to the expert's opinion, The original text was modified as follows(The modified parts are highlighted in yellow in the paper L216-L219)
“the mean values of MFCC eigenvalues of working sound of coal mining machine, road-header, and ventilator are have a significant difference in size in the first 6 dimensions of MFCC eigenvalues, and the remaining 6 dimensions do not differ greatly in size.”
Point 8. The curves in the pictures in the manuscript overlap each other, making it difficult to distinguish the changes of different curves. There was no good response.
Response: According to the expert's opinion, The original pictures was replaced as following (The replaced pictures are highlighted in yellow in the paper L158, L177, L224, L236, L247, L334)
Point 9. In the algorithm section, it is recommended to use the BP genetic algorithm to train MLP. Where are the improvements? There was no good response and further explanation was needed from the author.
Response: The paper use supervised learning of the BP genetic algorithm to train MLP, the improved parts are: (1) making the learning process of deep learning networks supervised. (2) The process of feature extraction is unsupervised, proposed Relief algorithm to construct the optimal feature selection model construction, complete the independent selection of sound features.
Point 10.The conclusion part proposes that it can effectively distinguish between coal mine gas explosion sound and non-coal mine gas explosion sound. But there is no distinction in the text between the sound of a coal mine explosion and the sound of a non-coal mine explosion? There was no good response and further explanation was needed from the author.
Response: In section 2.2, the paper analyzed 70-dimensional features of gas and coal dust explosion sounds and non-gas and coal dust explosion sounds, which include statistical features, time-domain features, and frequency-domain features. To visualize the difference between the sound of a coal mine explosion and the sound of a non-coal mine explosion, the paper took the gas explosion sound, coal dust explosion sound, coal mining machine working sound, roadheader working sound and ventilator working sound as the research objects, analyzed their 70-dimensional sound characteristics, and the RELIEF algorithm was proposed to achieve the selection of the best features.

Reviewer 2 Report
The authors have made revisions and additions based on the reviewers' comments. However, there seems still some sections that need to be revised.
Discussions on the comparison of amplitude of sound signal features are still generally qualitative and ambiguous, with expressions such as "not significant" and "small difference". It would not be difficult to make quantitative comparisons using such as dB to current data.
L84; (Additional remarks); Why are 2.1 Sound material and 2.2 Sound material in the same title?
L138 equation 7; variable R should be explained.
L154; the mean values of the spectral roll-off and the over-zero rate differed considerably, and the mean values of the spectral center of mass and the spectral diffusion did not differ much.;
Match the terms spectral roll-off, over-zero rate and spectral diffusion with the terminology used in Figure 2.
L357 Equation 11; (y-y)^2; I do not see a hat of y in the equation, it is (y-y)^2 (=0).
Shouldn't the variable p(1 to M) in the Sigma symbol be a subscript of y? But I don't see any subscript for y.
Author Response
Point 1. L84; (Additional remarks); Why are 2.1 Sound material and 2.2 Sound material in the same title?
Response: According to the expert's opinion, The original text was modified as follows(The modified parts are highlighted in yellow in the paper L92)
“2.2. Sound characteristics”
Point 2. L138 equation 7; variable R should be explained.
Response: According to the expert's opinion, The original text was added as follows(The added parts are highlighted in yellow in the paper L140):
“R is the length of the sound signal.”
Point 3. L154; the mean values of the spectral roll-off and the over-zero rate differed considerably, and the mean values of the spectral center of mass and the spectral diffusion did not differ much.;Match the terms spectral roll-off, over-zero rate and spectral diffusion with the terminology used in Figure 2.
Response: According to the expert's opinion, The original text was modified as follows(The modified parts are highlighted in yellow in the paper L154-L156)
“the mean values of roll-off coefficient and zero crossing rate differed considerably, and the mean values of spectral center of mass and spectral spread did not differ much.”
Point 4. L357 Equation 11; (y-y)^2; I do not see a hat of y in the equation, it is (y-y)^2 (=0).
Shouldn't the variable p(1 to M) in the Sigma symbol be a subscript of y? But I don't see any subscript for y.
Response: According to the expert's opinion, The original text was modified as follows(The modified parts are highlighted in yellow in the paper L358-L359)
“ (11)
Where is the true label and is the predicted value.”

Round 3
Reviewer 1 Report
This paper mainly writes about the MLP-based coal mine gas coal dust explosion sound recognition method, which can be received after modification, please enrich it again.